# Polyphenols, Antioxidant Activity and Volatile Compounds in Fermented Leaves of Medicinal Plant Rosebay Willowherb (*Chamerion angustifolium* (L.) Holub)

**DOI:** 10.3390/plants9121683

**Published:** 2020-12-01

**Authors:** Elvyra Jariene, Marius Lasinskas, Honorata Danilcenko, Nijole Vaitkeviciene, Alvyra Slepetiene, Katarzyna Najman, Ewelina Hallmann

**Affiliations:** 1Agriculture Academy, Institute of Agriculture and Food Sciences, Vytautas Magnus University, Donelaicio str. 58, 44248 Kaunas, Lithuania; elvyra.jariene@vdu.lt (E.J.); honorata.danilcenko@vdu.lt (H.D.); nijole.vaitkeviciene@vdu.lt (N.V.); 2Institute of Agriculture, Lithuanian Research Centre for Agriculture and Forestry, Instituto al. 1, Akademija, LT-58344 Kedainiai, Lithuania; alvyra.slepetiene@lammc.lt; 3Department of Functional and Organic Food, Institute of Human Nutrition Sciences, Warsaw University of Life Sciences, Nowoursynowska 15c, 02-776 Warsaw, Poland; katarzyna_najman@sggw.edu.pl (K.N.); ewelina_hallmann@sggw.edu.pl (E.H.)

**Keywords:** flavonoids, phenolic acids, solid state fermentation, odors

## Abstract

At present, the consumption of medical plants and functional foods is growing in the whole world. Rosebay willowherb (*Chamerion*
*angustifolium* (L.) Holub) is an important medicinal plant that has various pharmacological effects (antioxidant, anti-inflammatory, anticancer, and others), can improve the state of health and well-being, and reduce the risk of various diseases. The aim of this work was to investigate volatile compounds, polyphenols, and antioxidant activity in rosebay willowherb leaves fermented for 24 and 48 h in solid state fermentation under aerobic and anaerobic conditions. High-performance liquid chromatography (HPLC) for polyphenols and the spectrophotometric method for antioxidant activity determinations were used. To recognize and identify the leaves’ fragrances, electronic nose (Alpha M.O.S) measurement technology was used. The results showed that the highest amounts of total polyphenols in dried matter were after 48 h aerobic solid state fermentation (SSF). Antioxidant activity was higher under 48 h SSF compared to the control. The most abundant flavoring compound groups were esters, terpenes, and aldehydes. In unfermented leaves, (z)-3-hexen-1-ol, acetate, hexyl acetate, and trans-hex-2-enyl acetate prevailed, characterized by fragrances of greenery, flowers, and fruits. The undesired esters group compounds, ethyl butyrate and butyl acetate, with pungent odor, were detected after 48 h anaerobic SSF.

## 1. Introduction

Plants, as sedentary organisms, have to adjust to the surrounding environment during their life cycle. To compensate for their immobility, plants have evolved various mechanisms for their interactions with the environment, including the release of arrays of volatile compounds from their leaves, flowers, and fruits into the atmosphere and from roots into the soil. At present, a total of 1700 volatile compounds have been described from more than 90 plant families. Because of the volatile compounds and some other bioactive substances, e.g., phenolics, many plant species have found their use as medicinal plants [1].

The rosebay willowherb (*Chamerion angustifolium* (L.) Holub) takes a place in traditional and folk medicine, and is increasingly used in the diet for humans, due to its high antioxidant capacity and therapeutic properties [2,3]. This medicinal plant is a source of polyphenolic substances found mainly in flavonoids, phenolic acids, and hydrolyzed tannins (ellagitannins) [4]. In folk medicine, these herbs are known for their anti-inflammatory, analgesic, and antispasmodic effects, for the treatment of benign prostatic hyperplasia, and for the prevention of urinary problems [5,6,7].

In order to fully reveal the potential of the rosebay willowherb and give it a higher chemical value, the leaves of this plant have been fermented in solid state fermentation (SSF) (a fermentation method when no additional water is used) for a long time. It is expected that cutting and pressing during SSF could intensify cell wall degradation, thus improving diffusion of the bioactive compounds from the inner cell parts, and then, initiating better extraction. In this way, the amounts of biologically active substances are modified and taste and odor-forming compounds are formed. Volatile compounds, such as polyphenols, polysaccharides, and alkaloids, play important roles in communication and defense in plants [8]. The aroma of plants is produced by a series of complex low-molecular-weight substances which are a class of lipophilic substances with an aromatic odor [9].

The sensory properties of food products are very important for consumers. The purpose of this work was to investigate polyphenols, antioxidant activity, and volatile compounds in fermented rosebay willowherb leaves under different conditions and to determine the influence of solid state fermentation on the sensory properties of the rosebay willowherb leaves under different conditions (24 and 48 h aerobic or anaerobic fermentation). In the future, these studies could lead to the development of rosebay willowherb leaves as high-value, functional, and healthy foods or medicines.

## 2. Results

### 2.1. The Amounts of Polyphenols Compounds

According to the obtained results, SSF changed the quantities of polyphenols in fermented rosebay willowherb leaves in aerobic and anaerobic conditions. The amounts of total polyphenols were higher after 24 h in aerobic and anaerobic (12.26% and 23.29%, respectively) and 48 h in aerobic and anaerobic (accordingly to 83.31% and 45.45%) SSF, compared to the control (unfermented). The quantities of separate biologically active substances differed depending on SSF conditions (Table 1).

The amounts of ellagitannin oenothein B significantly increased after 24 h in aerobic and in anaerobic (536.79 and 445.47 mg 100 g^−1^ DW, respectively), and after 48 h in anaerobic (527.87 mg 100 g^−1^ DW) SSF, compared to control. 

The content of total flavonoids compared with the control significantly increased in all tested variants: after 24 h in aerobic SSF—45.76%, after 24 h in anaerobic SSF—18.48%, after 48 h in aerobic SSF—30.50%, and after 48 in anaerobic SSF—54.93%. 

The quantities of individual flavonoids compounds quercetin-3-O-rutinoside and quercetin-3-O-glucoside increased the most under 48 h SSF (Table 1).

Total phenolic acids quantities increased after 24 h in anaerobic and 48 h in aerobic and in anaerobic SSF. Ellagic acid is the most abundant phenolic acid in rosebay willowherb leaves and after 48 h aerobic SSF, its amounts significantly increased from 1246.56 (unfermented) to 2588.25 mg 100 g^−1^ DW (Table 1).

### 2.2. Antioxidant Activity

We found that antioxidant activity in rosebay willowherb leaves significantly decreased after 24 h in aerobic SSF (19.23%) and after 24 h in anaerobic SSF (11.14%), but increased after 48 h in aerobic SSF (15.50%) and after 48 h in anaerobic SSF (14.27%) compared to unfermented leaves (324.56 mMTEAC/100 g DW) (Figure 1).

### 2.3. Volatile Organic Compounds in Rosebay Willowherb Leaves

The detection of odors by an electronic nose is based on the release of compounds and their identification and quantification in a free space or product matrix. This instrumental detection method usually identifies unsuitable compounds that cause changes in odor and taste and are undesirable in the product. These undesirable compounds are determined in the presence of changes in the odor profile and in light of changes in product conditions or acting factors. In addition, the electronic nose is perfectly adapted for product authentication.

The obtained results (Table 2, Table 3, Table 4 and Table 5) showed that the main and most abundant groups of flavoring compounds identified were esters, terpenes, and aldehydes.

From the group of esters, the highest peaks in unfermented rosebay willowherb leaves are at 59 retention times per second. Z-3-hexen-1-ol, acetate, hexyl acetate, and trans-hex-2-enyl acetate were found to be predominant in intensity (Table 2). These compounds were distinguished by fragrance of greenery, flowers, and fruits. Ethyl butyrate and butyl acetate, which have a pungent odor, were detected in the undesired esters group after 48 h of anaerobic fermentation.

The group of terpenes is the first in terms of the amount of volatile compounds found in the leaves of rosebay willowherb, but the most varied among the studied variants. From the variety of compounds of the terpenes group, citronellyl was determined in the leaves fermented under aerobic conditions at 24 and 48 h, according to the height of the peak (Table 3). It is a compound with a characteristic leafy odor. This compound was not observed in unfermented leaves, so it could be that its occurrence may have been induced by solid state fermentation. Furthermore, the compounds that were in unfermented leaves at the beginning of the fermentation process were not further detected.

Aldehyde hexanal, which is characterized by the smell of fresh grass and apples, was characteristic of all variants (Table 4). Phenylacetaldehyde was detected in unfermented and 24 h aerobic and anaerobic fermented rosebay willowherb leaves. 

In the 48-h aerobic SSF sample, we observed that the peaks of the identified compounds are best separated. After 48 h of aerobic SSF, isobutanol, an alcohol group compound with a reliability of more than 85%, was detected on the leaves of the rosebay willowherb in a non-polar column at 23 s of retention and at 33 s on a polar column (Table 5). Thus, the predominance of the alcohols group compound indicates that the higher alcohols that can be formed during prolonged SSF, and due to the wild yeast on the raw material, provide the undesirable taste and aroma.

Butanone, a ketones group compound that provides a pleasant fruity aroma, was found in all fermented variants, except the unfermented material. In addition, it was observed that in terms of percentage of relevance, the ketone 2, 3—pentanodion was detected in our test samples and its odor is characteristic of black tea.

Strong and very unpleasant odors are provided by sulfur-containing compounds. Organic sulfur compounds (also known as mercaptans), such as dimethyl sulfide and dimethyl trisulfide, have the characteristic smell of rotten vegetables. These compounds, based on the retention time of 15–20 s, were detected in all test variants.

## 3. Discussion

According to our earlier and current research results, to improve the extraction of biologically active substances from rosebay willowherb leaves, solid state fermentation (SSF) could be used [10].

Microbial metabolism and enzymes produced during solid state fermentation, such as polyphenol oxidase, have a significant effect on the components of fermented rosebay willowherb leaves, by breaking down macromolecular components (proteins, lipids, and polysaccharides) into lower molecular weight substances and secondary products of metabolism [11]. Furthermore, cutting and pressing during SSF could intensify cell wall degradation, thus improving the diffusion of biologically active substances from the inner cell parts, and then, initiating better extraction.

Thus, it could have been one of the main factors in monitoring higher levels of polyphenols and antioxidant activity after SSF. However, there is no unanimous consensus on how the SSF process can influence the level of bioactive compounds. The obtained data showed that the SSF process increased the level of polyphenols. Kosman et al. 2013 confirmed that phenomenon [12]. However, different results were obtained by Kauppinen and Galambosi (2016), showing the opposite [13]. 

Flavonoids were found as the prime group of naturally occurring phenolic compounds [14]. In our experiment, the fermentation process increased the quantity of total flavonoids, but changed their proportional composition. As reported by Hallmann et al. (2017) [15], significantly lower total flavonoids content was found in fermented white cabbage juice (sauerkraut) compared to fresh juice. 

According to some researchers, oenothein B is among the compounds considered to be the primary biologically active components in rosebay willowherb extracts [16]. Our data confirmed that oenothein B was a prevailing compound among the polyphenols identified in rosebay willowherb.

Changes in compounds concentrations and profiles occur in fresh tea leaves as well as during tea processing [17,18]. With the help of an electronic nose, certain compounds that form complexes of odors can be identified. Aroma is usually described by the chemical structure, functional groups, such as heterocyclic systems, double bonds, or aromatic rings of the compounds that make it up, which contribute to the overall molecular form and cause a certain odor [19].

The use of microorganisms in the food industry, and especially in fermentation processes, is inseparable from the development of new flavors and aromas. The literature data on the composition of volatile compounds in the rosebay willowherb are scarce, but according to scientists, the main classes of aromatic compounds in the fresh leaves of *Chamerion angustifolium* contain terpenes and esters. While phenolic compounds are responsible for the taste, volatile compounds are fundamental for tea aroma [20,21]. The aromatics identified in the rosebay willowherb leaves include different classes of chemical compounds such as: alcohols, aldehydes, esters, terpenes, organic acids, pyrazines, ethers, ketones, sulfides.

The chromatograms of the identified volatiles are shown in Appendix A:In unfermented leaves—117 peaks (Figure A1)In 24 h fermented leaves—under aerobic conditions (103 peaks) (Figure A2), anaerobic (119 peaks) (Figure A3);In 48 h fermented—aerobic conditions (105 peaks) (Figure A4), anaerobic (122 peaks) (Figure A5)

Depending on the solid state fermentation conditions, 57 major compounds with intense odor were identified in rosebay willowherb leaves, of which there were 37 compounds in unfermented leaves, 28 compounds in 24-h aerobic conditions, 33 compounds in 24-h anaerobic conditions, 27 compounds in 48-h aerobic conditions, and 29 compounds in 48-h anaerobic fermented leaves.

The results showed that solid state fermentation led to the formation of some new volatile compounds in the fermented leaves of rosebay willowherb. For example, citronellyl (terpenes group), isobutanol (alcohols group), and butanone (ketones group) appeared only after SSF. This could be explained because microorganisms and intercellular enzymes change the composition of products during fermentation, causing the breakdown of macromolecular substances into smaller and at the same time, produce metabolic products (acids, alcohols, esters, aldehydes, ketones, etc.) [22]. However, according to Catunescu et al. [23], water loss and tissue degradation can reduce the amounts of volatile compounds in the raw material, which was also observed in this experiment after solid state fermentation. 

The main established odor profiles of unfermented rosebay willowherb leaves themselves presented pungent odor and that of greenery with aromas of fruits. After fermentation, aromas of fruits, such as lemons and apples, were further revealed and compounds characteristic of this odor were formed.

## 4. Materials and Methods 

### 4.1. Chemicals

ABTS (2,2′-azino-bis(3-ethylbenzothiazoline-6-sulfonic acid) diammonium salt (Sigma-Aldrich, Poland); acetonitrile at HPLC purity (Sigma-Aldrich, Poznan, Poland); deionized water (Sigma-Aldrich, Poland); methanol at HPLC purity (Merck, Poznan, Poland); ortho-phosphoric acid 99.9% (Chempur, Piekary Śląskie, Poland); phenolics standards (purity 99.5–99.9%) of gallic, chlorogenic, *p*-coumaric benzoic, ellagic, oenothein B, quercetin-3-*O*-rutinoside, myricetin, luteolin, quercetin, quercetin-3-*O*-glucoside, and kaempferol (Sigma-Aldrich, Poland); and phosphate-buffered saline (Merck, Poznan, Poland) were used.

### 4.2. Object, Time, and Place of Research

The leaves of *Chamerion angustifolium* (L.) Holub were selected as the object of research. The raw material of the rosebay willowherb was collected during mass flowering in the first week of July 2019 in Safarka village, Jonava district, Giedres Nacevicienes organic farm (Lithuania, latitude, 55°00′22″ N; longitude, 24°12′22″ E).

### 4.3. Methods for Determination of Quantitative and Qualitative Physical Parameters of Rosebay Willowherb Leaves

Rosebay willowherb leaves were collected at random from the experimental site. The total sample of leaves was 420 g. For the solid state fermentation process, the raw material was comminuted with plastic knives. For laboratory tests, the sample of leaves was divided as follows:-100 g for control unfermented leaves;-160 g for 24 h and 48 h for aerobic solid state fermentation;-160 g for 24 and 48 h anaerobic solid state fermentation.

The raw material prepared for the fermentation process was divided into 80 g and placed in sterile jars. The leaves were compressed in jars in such a way as to release the moisture required for the solid state fermentation process. The jars for the aerobic process were covered with sterile merle, and the jars for the anaerobic process were sealed with lids.

The whole solid state fermentation process took place in a dark room at 30 °C. The temperature was recorded at the sample stand before the start of the process, once in the first 24 h, and at the end of the fermentation. The temperature was measured 3 times during the 24-h fermentation process and 5 times during the 48-h fermentation process. 

For polyphenols and antioxidant activity, the raw leaves were dried at 40 °C for 10 h in a Termaks drying oven (Bergen, Norway) until constant weight and leaves were milled in a knife mill Grindomix GM 200 (Retsch GmbH, Haan, Germany) and kept in closed containers at a temperature of 25 °C in the ventilated, dry, dark, and cool room. All chemical experimentations were performed three times.

The odor of rosebay willowherb leaves was assessed using an electronic nose (Alpha M.O.S). Samples were formed in airtight tubes to determine the odor of rosebay willowherb leaves. A total of 1.5 g of unfermented (control) raw material was added to the tubes, and 1 g after the solid state 24 and 48 h fermentation process. Three replicates were performed to ensure the accuracy of the results. The electronic nose parameters were set to heat the analyzed sample to 50 ° C and heat for 87 to 90 s. The temperature of the valve was set at 250 °C and that of the injection device (injector) at 200 °C. Analysis (incubation) of each tube lasted 900 s. A total of 3.5 mL of odor from each tube was added for analysis of the odor profile. With the help of the electronic nose (Alpha M.O.S), organic volatile compounds, which formed an odor profile, were identified. The analysis of the compounds was based on the calculated Kovats index (retention) and mass spectra for each peak of the chromatogram were analyzed. Two types of columns—non-polar column (MXT-5) and polar column (MXT-1701)—were used. Kovats indexes were calculated according to the formula:(1)$$I^{T}=100\left[\frac{t_{Ri}^{T}-t_{Rz}^{T}}{t_{R\left(z+1\right)}^{T}-t_{Rz}^{T}}+z\right]$$
where $t_{Ri}^{T}$—retention time of the sample peak; $t_{Rz}^{T}$—retention time of the n-alkane peaks upstream of the sample peak; $t_{R\left(z+1\right)}^{T}$—retention time of n-alkane peaks immediately after the sample peak; *z*—is the number of carbon atoms in the n-alkane peak before the sample peak.

### 4.4. Measurement of Polyphenols

For the determination of polyphenols, the high-performance liquid chromatography (HPLC) method described by Hallmann et al. (2019) was used [24]. A total of 100 mg powder of dried rosebay willowherb leaves was mixed with 5 mL of 80% methanol (80:20 methanol and ultrapure water), and then, plastic tubes were closed by plastic cap and shaken with a vortex (60 s). Then, in an ultrasonic bath, all examples were extracted for 10 min, at a temperature of 30 °C, and 5.5 kHz. After 15 min, the samples were centrifuged for 10 min, 3780× *g*, and temperature was 5 °C. A clean plastic tube was used to collect the supernatant, and then, it was centrifuged once more for 5 min, 31,180× *g*, at 0 °C temperature. Supernatant (850 μL) was moved to a vial (HPLC) and was analyzed. For polyphenols separation, a Synergi Fusion-RP 80i Phenomenex column (250 × 4.60 mm) was used. Shimadzu equipment (two pumps (LC-20AD), a controller (CBM-20A), a column oven (SIL-20AC), and UV–vis spectrometer (SPD-20 AV)) was used to carry out the analysis. The phenolics were isolated with these gradient conditions: a flow rate of 1 mL min^−1^; we used two gradient phases—10% (*v*/*v*) acetonitrile and ultrapure water (phase A) and 55% (*v*/*v*) acetonitrile and ultrapure water (phase B). Orthophosphoric acid was used to acidify the phases (pH 3.0). The whole analysis lasted 38 min. The program of the stage was: 1.00–22.99 min, 95% phase A and 5% phase B; 23.00–27.99 min, 50% phase A and 50% phase B; 28.00–28.99 min, 80% phase A and 20% phase B; and 29.00–38.00 min, 95% phase A and 5% phase B. For flavonoids, wavelengths were 250 nm, and for phenolic acids, 370 nm. The pure standards 99.9% (Poland, Sigma-Aldrich) were used to identify the polyphenols.

### 4.5. Antioxidant Activity

Srednicka-Tober et al. described the method for antioxidant activity determination [25]. The sample of 250 mg dried plant powder was placed into a plastic tube, and distilled water was added (25 mL). A vortex was used for 1 min to mix the samples (Labo Plus, Warsaw, Poland). In the next stage, the samples were placed into a shaker incubator (IKA, Staufen im Breisgau, Germany) for 1 h, at 30 °C. Then, the samples were shaken again and centrifuged (Centrifuge, MPW-380 R, Warsaw, Poland) for 15 min, 14,560× *g,* at temperature 5 °C. After that, the supernatant was gathered for measurements. In laboratory glass tubes, the samples were determined with a measurement dilution scheme (0.5–1.5 mL) and then, placed in 3.0 mL of ABTS·+ cationic solution in phosphate-buffered saline (PBS). After 6 min, a spectrophotometer (Helios γ, Thermo Scientific, Warsaw, Poland) was used to take the samples’ absorbances (21 °C, wavelength λ = 734 nm). The received calculations were measured with a special formula and also, the dilution factor was used. The conclusive data were expressed in mmol of TE (Trolox equivalents/100 g dry weight (DW)).

### 4.6. Mathematical-Statistical Analysis of Research Data

The ANOVA (a two-way analysis of variance) method, with software package STATISTICA, was used to perform statistical analysis of the obtained data (Statistica 12; StatSoft, Inc., Tulsa, OK, USA). The results were submitted as the mean with standard error. The Fisher’s LSD test was used to estimate the statistical significance of differences between the means (*p* < 0.05). To recognize and identify leaves fragrances, electronic nose (Alpha M.O.S) measurement technology was used.

## 5. Conclusions

This study showed that solid state fermentation (SSF) has a high potential to modify biologically active substances and to improve the aroma of rosebay willowherb leaves due to the release of volatile compounds and changes in the esters, terpenes, and aldehydes biosynthesis. 

The results showed that the amounts of total polyphenols were higher after 24 and 48 h in aerobic and anaerobic SSF. Antioxidant activity in rosebay willowherb leaves significantly increased after 48 h in aerobic and in anaerobic SSF. The aroma profiles of unfermented rosebay willowherb leaves had greenery, acid, and fruity-like odors, but after fermentation, apples and citrus-like odors prevailed. 

Depending on the SSF conditions, 57 main compounds with intense odor were identified: 37 compounds in unfermented leaves, 28 compounds after 24 h SSF under aerobic conditions, 33 compounds after 24 h SSF under anaerobic conditions, 27 compounds after 48 h SSF under aerobic conditions, and 29 compounds after 48 h SSF under anaerobic conditions. According to intensity, in unfermented leaves, (z)-3-hexen-1-ol, acetate, hexyl acetate, and trans-hex-2-enyl acetate prevailed, characterized by fragrances of greenery, flowers, and fruits. The undesired esters group compounds, ethyl butyrate and butyl acetate, which had a pungent odor, were detected after 48 h anaerobic fermentation. 

According to the obtained results, 48 h aerobic solid state fermentation could be a good choice to produce pleasant odor, rich with polyphenols and antioxidant activity, fermented rosebay willowherb leaves tea or food supplements.

## Figures and Tables

**Figure 1 plants-09-01683-f001:**
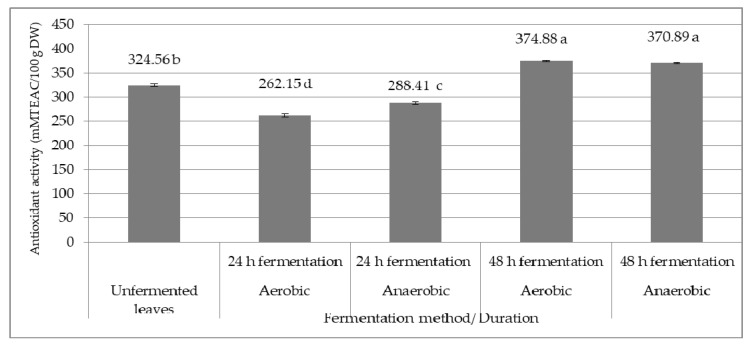
Antioxidant activity of rosebay willowherb leaves (*p* < 0.0001). Means followed by the different letters are significantly different at *p* < 0.05, *n* = 3.

**Table 1 plants-09-01683-t001:** The amounts of polyphenol compounds (mg 100 g^−1^ DW) in rosebay willowherb leaves as influenced by solid state fermentation conditions.

Polyphenol Compounds	Unfermented Leaves	24 h Fermentation		48 h Fermentation		*p*-Values (Fermentation Duration x SSF ^2^ Method)
		Aerobic	Anaerobic	Aerobic	Anaerobic	
Oenothein B	371.13 ± 40.94 c ^1^	536.79 ± 2.62 a	445.47 ± 8.47 b	387.22 ± 56.36 c	527.87 ± 5.43 a	<0.0001
Quercetin-3-O- rutinoside	25.21 ± 2.71 d	37.24 ± 0.81 c	32.36 ± 0.90 c	64.14 ± 1.18 b	97.42 ± 5.83 a	<0.0001
Myricetin	11.31 ± 1.70 c	8.71 ± 0.11 d	8.29 ± 0.03 d	25.83 ± 0.26 a	14.98 ± 1.02 b	<0.0001
Luteolin	4.85 ± 0.06 a	4.84 ± 0.20 a	2.29 ± 0.07 b	4.90 ± 0.08 a	2.25 ± 0.11 b	ns
Quercetin	2.83 ± 0.21 cd	3.82 ± 0.32 b	2.54 ± 0.13 d	2.99 ± 0.01 c	6.30 ± 0.29 a	<0.0001
Quercetin -3-O- glucoside	13.10 ± 1.39 c	34.05 ± 1.95 b	14.44 ± 1.75 c	73.98± 0.24 a	16.01 ± 2.24 c	<0.0001
Kaempferol	2.43 ± 0.05 c	2.58 ± 0.03 c	5.09 ± 0.32 a	3.18± 0.02 b	2.69 ± 0.07 c	<0.0001
TF ^3^	430.85 ± 43.69 c	628.02 ± 3.77 a	510.49 ± 7.69 b	562.24 ± 57.57 b	667.52 ± 8.32 a	<0.0002
Gallic acid	6.94 ± 0.51 c	16.84 ± 0.35 b	6.02 ± 0.48 c	7.00 ± 0.63 c	23.25 ± 3.06 a	<0.0001
Chlorogenic acid	5.13 ± 0.40 d	7.30 ± 0.62 c	8.08 ± 0.56 c	23.74 ± 1.55 a	11.98 ± 0.52 b	<0.0001
*P*-coumaric acid	67.55 ± 9.28 c	61.13 ± 2.21 c	251.69 ± 7.72 b	59.93± 2.53 c	298.83 ± 8.91 a	<0.0001
Ellagic acid	1246.56 ± 10.71 c	1260.48 ± 26.50 c	1400.78 ± 333.98 bc	2588.25± 31.96 a	1562.00 ± 154.78 b	<0.0001
Benzoic acid	14.01 ± 0.42 a	14.40 ± 1.54 a	6.50 ± 0.45 c	5.38± 0.36 c	12.34 ± 0.40 b	<0.0001
TPA ^4^	1340.19 ± 7.61 c	1360.14 ± 30.55 c	1673.08 ± 327.65 b	2684.30 ± 31.33 a	1908.40 ± 148.64 b	<0.0002
TP ^5^	1771.04 ± 43.54 d	1988.16 ± 32.03 cd	2183.57 ± 332.05c	3246.54 ± 85.52 a	2575.91 ± 152.90 b	<0.0013

^1^ Means within rows followed by different letters are significantly different at *p* < 0.05. Average value ± standard error, *n* = 3. ^2^ SSF—solid state fermentation. ^3^ TF—total flavonoids. ^4^ TPA—total phenolic acids. ^5^ TP—total polyphenols. ns—not significant.

**Table 2 plants-09-01683-t002:** Influence of solid state fermentation conditions on the composition of the esters group volatile organic compounds in rosebay willowherb leaves.

Implied Compound	Non-Polar Columns (MXT-5) RT ^1^/KI ^2^—Polar Columns (MXT-1701) RT/KI					Odor Description ^3^
	Unfermented Leaves	24 h Fermentation		48 h Fermentation		
		Aerobic	Anaerobic	Aerobic	Anaerobic	
Esters						
Ethyl acetate	21.77/615 26.67/671	21.79/615 26.66/671	21.70/614 27.69/685	21.87/616 26.59/670	21.78/615 27.77/686	Sweet, grapes
Isopropyl acetate	-	-	25.41/662 31.77/728	25.43/662 31.85/729	-	Fruits, bananas
Ethyl butyrate	-	39.68/801 47.41/874	39.66/801 47.38/873	-	39.76/802 ^4^ 47.44/874^4^	Apples, pineapples, butter
Butyl acetate	-	39.68/801 47.41/874	39.66/801 47.38/873	-	39.76/802 ^4^ 47.44/874 ^4^	Apples, bananas, pungent
Ethyl hexanoate (>80)	59.01/998 ^4^ 63.67/1057 ^4^	-	-	-	-	Pineapples
Z-3-hexen-1-ol acetate (>95)	59.64/1007 ^4^ 65.49/1081^4^	59.53/1005 ^4^ 65.54/1082 ^4^	59.02/999 ^4^ 65.67/1084 ^4^	59.51/1005 ^4^ 65.53/1082 ^4^	59.06/999 65.63/1083	Bananas, flowers, greenery
Hexyl acetate (>90)	59.64/1007 ^4^ 65.49/1081 ^4^	59.53/1005 ^4^ 65.54/1082 ^4^	59.02/999 ^4^ 65.67/1084 ^4^	59.51/1005 ^4^ 65.53/1082 ^4^	59.06/999 ^4^ 65.63/1083 ^4^	Apples, bananas, grass
Trans-hex-2-enyl acetate	59.64/1007 ^4^ 65.49/1081 ^4^	59.53/1005 ^4^ 65.54/1082 ^4^	59.02/999 65.67/1084	59.51/1005 65.53/1082	59.74/1008 ^4^ 65.63/1083 ^4^	Greenery
Benzyl acetate	-	-	-	71.19/1177 77.64/1278	-	Fruits
Ethyl octanoate	-	-	-	72.03/1191 76.31/1245	-	Apricots, flowers, Pineapples
Phenethyl acetate	76.12/1265 81.34/1350	-	76.19/1266 82.10/1365	-	-	Flowers, honey, roses
δ-decalactone	-	87.02/1487 98.37/1726	-	87.97/1508 98.48/1728	87.22/1491 98.32/1724	Coconuts, butter

^1^ RT—retention time (min); ^2^ KI—Kovats index; ^3^ Sensory description from AroChemBase; ^4^ Volatile compounds are identified as relevant and have been identified as more than 75% reliable.

**Table 3 plants-09-01683-t003:** Influence of solid state fermentation conditions on the composition of volatile organic compounds of the terpenes group in rosebay willowherb leaves.

Implied Compound	Non-Polar Columns (MXT-5) RT ^1^/KI ^2^—Polar Columns (MXT-1701) RT/KI					Odor description ^3^
	Unfermented Leaves	24 h Fermentation		48 h Fermentation		
		Aerobic	Anaerobic	Aerobic	Anaerobic	
Terpenes						
1R-(+)-α-pinene (>90)	52.64/925 ^4^ 54.41/946 ^4^	-	52.69/926 53.88/940	-	52.78/927 53.94/941	Cedars, pines, pungent
1S-()- α -pinene (>80)	52.64/925 ^4^ 54.41/946 ^4^	-	52.69/926 53.88/940	-	-	Cedars, pines, pungent
α -felandrene	59.01/998 62.02/1035	60.10/1013 62.05/1036	-	-	-	Citrus, fruits, woods
α-terpinene	59.64/1007 ^4^ 65.49/1081^4^	59.53/1005 ^4^ 65.54/1082 ^4^	59.02/999 65.67/1084	59.51/1005 65.53/1082	59.06/999 65.63/1083	Lemons
Limonene	61.05/1026 63.67/1057	-	-	-	-	Oranges
L-limonene	61.05/1026 63.67/1057	-	-	-	-	Oranges
γ-terpinene (>80)	64.04/1067 ^4^ 65.49/1081 ^4^	62.81/1050 ^4^ 66.22/1091 ^4^	-	-	62,66/1048 65,63/1083	Bitter, citrus
Citronellyl	-	68.42/1133 ^4^ 76.16/1251 ^4^	69.37/1148 ^4^ 75.45/1238 ^4^	69.36/1148 ^4^ 75.58/1241 ^4^	68.50/1134 75.49/1239	Leafy, citrus
Terpinen-4-ol	-	-	-	71.19/1177 ^4^ 77.64/1278 ^4^	-	Nutmegs, earthy, woods
Citronellol	74,68/1238 ^4^ 80,51/1333 ^4^	74.74/1239 80.48/1332	74.77/1240 80.54/1334	74.84/1241 ^4^ 80.63/1335 ^4^	74.83/1241 80.48/1332	Citrus, greenery, roses
Geraniol	-	-	76.19/1266 82.10/1365	-	-	Lemons peels, peach, roses
S-Carvone	76.12/1265 83.47/1393	-	75.58/1255 83.50/1394	-	-	Basils, cumins, fennels, mints
L-Carvone	76.12/1265 83.47/1393	-	-	-	-	Basils, cumins, fennels, mints

^1^ RT—retention time (min); ^2^ KI—Kovats index; ^3^ Sensory description from AroChemBase; ^4^ Volatile compounds are identified as relevant and have been identified as more than 75% reliable.

**Table 4 plants-09-01683-t004:** Influence of solid state fermentation on the composition of volatile organic compounds of aldehydes group in rosebay willowherb leaves.

Implied Compound	Non-Polar Columns (MXT-5) RT ^1^/KI ^2^—Polar Columns (MXT-1701) RT/KI					Odor Description ^3^
	Unfermented Leaves	24 h Fermentation		48 h Fermentation		
		Aerobic	Anaerobic	Aerobic	Anaerobic	
Aldehyds						
Acetaldehyde	14.48/452 17.35/502	-	14.45/451 17.32/501	-	14.52/453 17.39/503	Flowers, apples
Propanal	14.48/452 20.19/568	-		-	-	Fruits
Butanal	-	-	20.14/589 27.69/685	-	-	Unpleasant, sharp
3-metilbutanal (>95)	-	-	24.70/653 ^4^ 31.77/728 ^4^	-	-	Malt
Hexanal	39.72/801 50.40/902	39.68/801 47.41/874	39.66/801 47.38/873	37.74/784 50.46/902	39.76/802 47.44/874	Fresh grass, apples
Benzaldehyde	-	55.38/957 66.22/1091	-	-	-	Bitter almonds, malt
Octanal (>80)	59.64/1007 ^4^ 65.49/1081 ^4^	59.53/1005 ^4^ 65.54/1082 ^4^	59,02/999 ^4^ 65.67/1084 ^4^	59,51/1005 ^4^ 65.53/1082 ^4^	59,06/999 ^4^ 65.63/1083 ^4^	Citrus, fruits
Phenylacetaldehyde (>80)	62.62/1048 ^4^ 73.04/1196 ^4^	62.45/1045 ^4^ 73.69/1207 ^4^	62.65/1048 ^4^ 72.42/1186 ^4^	-	61.61/1034 71.95/1179	Berries, roses, honey
N-nonanal (>80)	65.97/1094 73.04/1196	66.00/1094 ^4^ 73.07/1196 ^4^	66.03/1095 ^4^ 72.42/1186 ^4^	66.63/1104 72.44/1186	66.06/1095 ^4^ 72.43/1186 ^4^	Fats, flowers, greens
Decanal	-	-	-	72.63/1201 78.39/1291	-	Flowers, oranges peels
2-decanal (>80)	76.12/1265 81.34/1350	-	72.02/1191 78.23/1288	-	-	Fats, fishs, oranges
P-anisaldehyde	76.12/1265 85.85/1444	-	-	-	-	Sweet, floral, anises

^1^ RT—retention time (min); ^2^ KI—Kovats index; ^3^ Sensory description from AroChemBase; ^4^ Volatile compounds are identified as relevant and have been identified as more than 75% reliable.

**Table 5 plants-09-01683-t005:** Influence of solid state fermentation on the composition of volatile organic compounds of alcohols, pyrazines, ethers, ketones, organic acids, and sulphides groups in rosebay willowherb leaves.

Implied Compound	Non-Polar Columns (MXT-5) RT ^1^/KI ^2^—Polar Columns (MXT-1701) RT/KI					Odor Description ^3^
	Unfermented Leaves	24 h Fermentation		48 h Fermentation		
		Aerobic	Anaerobic	Aerobic	Anaerobic	
Alcohols						
Isobutanol (>85)	-	22.63/626 ^4^ 31.80/728 ^4^	22.60/626 ^4^ 31.77/728 ^4^	22.60/626 ^4^ 33.09/740 ^4^	22.67/627 ^4^ 31.85/729 ^4^	Apples, wine
N-butanol	-	-	-	25.43/662 37.41/781	25.53/664 37.40/781	Fruits
Propane-1,2-diol	34.72/757 54.41/946	-	-	-	-	Slightly sweet
1-Hexanol	46.18/861 58.40/990	46.18/861 58.40/990	46.16/860 58.46/990	46.23/861 58.46/990	-	Herbs, grasses, flowers
1-octanol (>80)	-	-	-	-	64.13/1069 ^4^ 71.95/1179 ^4^	Burnt, bitter almonds
**Pyrazines**						
Trimethyl pyrazine (>80)	59.64/1007 ^4^ 67.05/1102 ^4^	60.10/1013 67.06/1102	59.02/999 67.12/1103	60.18/1014 ^4^ 66.21/1091 ^4^	59.74/1008 ^4^ 65.63/1083 ^4^	Cocoa, ground
**Ethers**						
Anethole (>80)	77.43/1289 ^4^ 83.47/1393 ^4^	-	-	-	-	Anises
**Ketones**						
Diacetyl	-	18.78/556 28.42/695			18.83/557 28.45/695	Butter, vinegars
Butanone (>85->95)	-	20.17/590 ^4^ 28.42/695 ^4^	20.14/589 ^4^ 27.69/685 ^4^	20.19/590 ^4^ 28.45/695 ^4^	20.21/591 ^4^ 27.77/686 ^4^	Fragrant, fruits, pleasant
2,3-pentanodion (>80)	28.36/700 39.04/796	28.36/700 ^4^ 39.07/796 ^4^	28.33/700 ^4^ 39.01/796 ^4^	28.41/700 ^4^ 39.10/797 ^4^	28.42/701 39.09/797	Butter, black tea, fruits
Nonan-2-one (>90)	65.18/1083 71.81/1176	66.58/1103 73.07/1196	66.03/1095 ^4^ 72.42/1186 ^4^	66.63/1104 72.44/1186	66.06/1095 ^4^ 72.43/1186 ^4^	Fruits, greens, hot milk
2-undecanone (>80)	77.43/1289 83.47/1393	-	77.51/1290 ^4^ 83.29/1389 ^4^	-	-	Fresh, greens, oranges
Acetophenone	65.18/1083 73.04/1196	-	-	-	-	Almonds, flowers
**Organic acids**						
Vinegar acid	21.77/615 37.09/778	-	-	21.87/616 37.41/781	-	Acids, vinegars
2-methylpropane acid	-	37.12/778 57.40/979	-	37.06/778 57.50/980	-	Burnt, butter, cheese
Pentanoic acid	-	-	-	-	50.40/899 65.63/1083	Fruits
Hexanoic acid (>90)	58.19/989 ^4^ 72.37/1185 ^4^	58.16/989 73.07/1196	58.21/989 ^4^ 72.42/1186 ^4^	-	58.26/990 ^4^ 72.43/1186 ^4^	Oil, cheese
Benzoic acid (>80)	69.24/1146 ^4^ 79.67/1316 ^4^	-	-	-	-	Sharp, sour
**Sulphides**						
Dimethyl sulfide	15.49/476 20.19/568	15.48/476 20.20/568	15.45/475 20.17/567	15.51/477 20.22/568	15.52/477 20.24/569	Cabbages, wet ground
Dimethyl trisulfide	55.80/961 62.02/1035	55.38/957 62.05/1036	55.84/962 61.46/1028	55.45/957 62.13/1037	55.90/963 61.98/1035	Sulfuric, cabbage, onions

^1^ RT—retention time (min), ^2^ KI—Kovats index; ^3^ Sensory description from AroChemBase; ^4^ Volatile compounds are identified as relevant and have been identified as more than 75% reliable.

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
