# Peer review of "Polyphenols, Antioxidant Activity and Volatile Compounds in Fermented Leaves of Medicinal Plant Rosebay Willowherb (Chamerion angustifolium (L.) Holub)"

_plants, 2020, doi:10.3390/plants9121683_

Round 1

Reviewer 1 Report

In the introduction are paragraphs that are not relevant to the
content of the article.
The source of the reference standards and the rest of the chemicals used
in the study should be mentioned.
The geographical position of the locality where the plant was harvested must be specified. The method of extraction of bioactive compounds is not clearly described. In table no. 1, what does TF mean? Gallic acid is spelled correctly.
Only one method was used to evaluate the antioxidant activity.
For the results to be sustainable, at least two methods for antioxidant
activity should have been performed.
The references should be revised, respectively from the last 10 years.  

Reviewer 2 Report

Thank you for your work, a few comments enclosed in the maniscript

Reviewer 3 Report

The present manuscript entitled "Polyphenols, antioxidant activity and volatile compounds in fermented leaves of medicinal plant rosebay willowherb (Chamerion angustifolium (L.) Holub)"is of particular interest, based on the growing worldwide consumption of medical plants and functional foods.

The rosebay willowherb is however a plant that is still not studied in details. The authors focused their investigation on the polyphenols, antioxidant activity and volatile compounds in different conditions fermented rosebay willowherb leaves and to determine the influence of solid-state fermentation on the sensory properties of the rosebay willowherb leaves under different conditions (24 and 48 hours aerobic or anaerobic fermentation).

In addition, I have a few recommendations and remarks and following a minor revision I recommend the manuscript for publication due to possible benefits for the scientific community.

- Please include some information regarding the SSE in the introduction section

- Please, link the discussion of results to various references.

- Page 3q line 102 – technical error, please delete redundant bracket sign

- Page 12 line 255 – replace “ata” with “at a”

-Page 13 line 256 – replace “examples” with “samples”

- Page 13 line 289-296 – uniform formatting

Round 2

Reviewer 1 Report

The paper can be accepted for publication.
